# Multi-Omics Analysis Reveals the Gut Microbiota Characteristics of Diarrheal Piglets Treated with Gentamicin

**DOI:** 10.3390/antibiotics12091349

**Published:** 2023-08-22

**Authors:** Lijun Shang, Fengjuan Yang, Yushu Wei, Ziqi Dai, Qingyun Chen, Xiangfang Zeng, Shiyan Qiao, Haitao Yu

**Affiliations:** 1State Key Laboratory of Animal Nutrition and Feeding, Ministry of Agriculture and Rural Affairs Feed Industry Centre, China Agricultural University, Beijing 100193, China; shanglj1996@163.com (L.S.); yangfengjuan@cau.edu.cn (F.Y.); daiziqi@cau.edu.cn (Z.D.); chenqingyun1999@126.com (Q.C.); ziyangzxf@163.com (X.Z.); qiaoshiyan@cau.edu.cn (S.Q.); 2Beijing Bio-Feed Additives Key Laboratory, Beijing 100193, China; 3College of Animal Science and Veterinary Medicine, Shandong Agricultural University, Tai’an 271000, China; weiyushu@sdau.edu.cn

**Keywords:** gentamicin, ETEC, piglet, diarrhea, microbiota, metabolite

## Abstract

The involvement of alterations in gut microbiota composition due to the use of antibiotics has been widely observed. However, a clear picture of the influences of gentamicin, which is employed for the treatment of bacterial diarrhea in animal production, are largely unknown. Here, we addressed this problem using piglet models susceptible to enterotoxigenic *Escherichia coli* (ETEC) F4, which were treated with gentamicin. Gentamicin significantly alleviated diarrhea and intestinal injury. Through 16s RNS sequencing, it was found that gentamicin increased species richness but decreased community evenness. Additionally, clear clustering was observed between the gentamicin-treated group and the other groups. More importantly, with the establishment of a completely different microbial structure, a novel metabolite composition profile was formed. KEGG database annotation revealed that arachidonic acid metabolism and vancomycin resistance were the most significantly downregulated and upregulated pathways after gentamicin treatment, respectively. Meanwhile, we identified seven possible targets of gentamicin closely related to these two functional pathways through a comprehensive analysis. Taken together, these findings demonstrate that gentamicin therapy for diarrhea is associated with the downregulation of arachidonic acid metabolism. During this process, intestinal microbiota dysbiosis is induced, leading to increased levels of the vancomycin resistance pathway. An improved understanding of the roles of these processes will advance the conception and realization of new therapeutic and preventive strategies.

## 1. Introduction

Antibiotics are being used worldwide on a vast scale, including in humans, animals and agriculture [1]. Antibiotic therapies, although common in practice, may negatively impact health for various reasons. First, antibiotics can easily alter the gut microbiota structure, leading to dysbiosis [2,3,4], even in the initial stages of treatment [5]. The gut microbiota is a key factor for maintaining health and its dysbiosis is associated with a variety of host diseases, including but not limited to gastrointestinal diseases [6,7,8,9]. More importantly, any long-term antibiotic treatment strategy has the potential to induce resistance evolution [10,11]. In particular, in animal husbandry, the addition of therapeutic or sub-therapeutic doses of antibiotics for disease treatment and growth promotion has become a common practice, without rigorous testing. This creates favorable conditions for the generation of antibiotic-resistant bacteria in hosts and in the environment [12,13,14,15,16]. These characteristics highlight the fact that novel antibiotic-based treatment strategies that do not disturb resident microbiota or reduce the induction of resistance evolution are needed. Consequently, understanding the effect of different antibiotics is of practical importance, because microbiota modulation using antibiotics may be a therapeutic option with the potential to advance the new therapeutic and preventive strategies.

The enterotoxigenic *Escherichia coli* (ETEC) is the major etiological agent of diarrhea in newborn and weaned piglets. ETEC-associated disease results in significant costs to the global pig industry due to high morbidity and mortality, substantial veterinary and labor costs and growth retardation. Gentamicin was used as an oral medicine for the treatment of bacterial diarrhea [17,18]. However, it remains unclear whether the gut microbiota changes during gentamicin therapy following ETEC infection and whether there are potential effects of the gut microbiota in shaping host responses by altering metabolic profiles.

Here, F4-susceptible piglets were selected to establish an ETEC infection model. The results revealed that gentamicin significantly alleviated diarrhea and intestinal damage caused by ETEC infection, especially in the jejunum. At the same time, the evenness and structure of the jejunal microbiota were significantly decreased and changed. However, the ileal microbial diversity and structure did not change significantly. Similarly, the jejunal metabolites in the gentamicin-treated group were completely different from those in the other two groups, whereas the ileal metabolites were not significantly different among the three groups. Furthermore, we found that arachidonic acid metabolism and vancomycin resistance were the most significantly downregulated and upregulated pathways, respectively. Seven genera were found to be significantly associated with altered metabolite pathways, revealing a link between microbiota structure altered by gentamicin and metabolite function. It is hoped that by improving our understanding of the functional changes mediated by gentamicin and the related microbial structural alterations, new therapeutic and preventive strategies will be conceived and realized.

## 2. Results

### 2.1. Effects of Gentamicin on ETEC Infection in Piglets

Prior to the trial, 36 piglets with F4 susceptibility, half male and half female, were selected from a total of 147 piglets. Based on the pre-experiment, we chose a 100 mL, 5 × 10^8^ CFU/mL dose to build the model with moderate symptoms. The diarrhea and inflammation in this model persisted for at least 5 days, which covered the entire trial period.

ETEC infection promotes small intestinal lesions and alters intestinal permeability, disrupting absorption and secretion, in turn resulting in diarrhea and decreased growth performance [19,20,21,22]. Given that ETEC infection causes the above characteristics, we assessed growth performance, diarrhea, and small intestinal pathology after gentamicin treatment. After gentamicin treatment, the growth performance of the infected piglets was measurably improved, although not significantly (Appendix A). Similarly, a significant improvement was observed in the fecal fluid content and diarrhea score (Figure 1B,C). A histological analysis showed obvious impairment in the jejunum of the ETEC-challenged piglets, as reflected by their severely damaged mucosal structures and substantial inflammatory cell infiltration (Figure 1D). In contrast, the ETEC + Gen group showed an intact mucosal layer and a clear and normal tissue structure (Figure 1D). At the same time, an increased crypt depth and decreased ratio of villus height to crypt depth (VCR) were observed in the jejunum, and a decreased villous height was observed in the ileum after ETEC challenge; however, the ETEC + Gen group showed minimal remission (Figure 1E). These results confirm the repair effect of Gen on intestinal infection. The serum enzyme activity measurements of DAO confirmed that Gen exhibited decreased intestinal permeability function (Figure 2A). In terms of circulating cytokines in the serum, a reduction in the levels of IL-1β and TNF-α was observed in the ETEC + Gen group compared to the ETEC group (Figure 2A). These results revealed the anti-ETEC infective effect of gentamicin on the piglets.

In addition, the ETEC challenge significantly increased the fecal *E. coli* populations but had little effect on the total number of bacteria (Figure 2B). Gentamicin treatment had no significant effect on these indexes, which was unexpected. This may be due to the short duration of gentamicin administration.

### 2.2. Diversity Analysis of Intestinal Microbiota

Through 16s RNA sequencing, we found that both the Ace and Simpson indexes in the jejunum were significantly increased in the ETEC + Gen group, which reflects increased species richness and decreased evenness compared with the other groups (Figure 3A). In contrast, there was no such effect on the ileum (Figure 3A). To further estimate the relatedness of the microbial communities among the groups, we calculated the distances between samples using Bray_Curitis (Figure 3B). Similar to the alpha diversity results, the principal coordinate analysis (PCoA) revealed clear clustering between the ETEC + Gen group and the other groups in the jejunum but did not reveal differences in microbial communities in the ileum between the groups (Figure 3B).

### 2.3. Gentamicin Treatment Altered Metabolite Levels

Since metabolites are the final result of microbial actions [23], we sought to provide deeper insights into the metabolic changes in the active fraction of the microbiota through metabolite profiling. Consistent with the microbial results, the principal component analysis (PCA) of metabolite profiling showed that the Gen group had a significantly different metabolite composition from the other two groups (Figure 3C). A comparison with the Kyoto Encyclopedia of Genes and Genomes (KEGG) compound database showed that the vancomycin resistance pathway and arachidonic acid metabolism pathway were the most significantly upregulated and downregulated pathways, respectively (Appendix A).

### 2.4. Correlation Analysis between Specific Metabolites and Microbiota

To further identify the key microorganisms in the ETEC + Gen group, correlation analysis was used. Five significantly altered metabolites in the vancomycin resistance pathway and nine significantly altered metabolites in the arachidonic acid metabolism pathway were used as environmental factors for an association analysis with microorganisms. Specifically, *Bacteroides*, *uncultured_bacterium_f_Lachnospiraceae*, *uncultured_bacterium_f_Muribaculaceae*, *Caldicoprobacter*, *Haloplasma*, *Caproiciproducens*, *Fastidiosipila*, *uncultured_bacterium_o_MBA03*, *Aminobacterium*, *Sedimentibacter* and *Lactobacillus* were significantly correlated with one or more metabolites (Figure 4). Further screening revealed that 7 of the 11 genera mentioned above showed significant differences between treatments (Figure 5). Specifically, *Bacteroides*, *uncultured_bacterium_f_Lachnospiraceae* and *uncultured_bacterium_f_Muribaculaceae* were positively correlated with metabolites in the vancomycin resistance pathway; *Caldicoprobacter*, *Caproiciproducens*, *Fastidiosipila* and *Haloplasma* were positively correlated with metabolites in the arachidonic acid metabolism pathway and/or negatively correlated with metabolites in the vancomycin resistance pathway.

## 3. Discussion

Antibiotics are widely used in human medicine and animal production, making important contributions to human health and animal husbandry development [24]. Today, however, the potential significance of the damaging effects of antibiotics on the gut microbiota has become a high-profile topic. Each use of antibiotics creates evolutionary pressure, both in human medicine and in veterinary medicine, leading to the emergence of resistance that poses a significant threat to public health security [25]. Due to the multiple levels of direct and indirect contact between humans and animals, this threat is compounded by the large degree of bacterial and antimicrobial gene exchange between humans and animals [26]. Consequently, a more comprehensive understanding of the impact of antibiotics on animal husbandry is needed. Here, we treated ETEC-infected piglets with gentamicin to address this issue.

ETEC challenge can cause impaired growth performance, an increased diarrhea rate, and small intestinal damage in piglets [27]. Although there was no significant improvement in growth performance, consistent with other findings on the use of in-feed antibiotics in previous studies, gentamicin treatment significantly alleviated the symptoms of diarrhea and intestinal injury [28,29,30]. These findings were also verified by significantly decreased DAO, IL-1β and TNF-α levels in the serum. The observed failure to significantly improve growth performance may be due to the inadequate duration of treatment in the present study.

Unsurprisingly, gentamicin caused significant changes in the diversity and structure of the jejunal microbiota. Our species diversity results, however, contradict other studies that have reported a decreased alpha diversity after antibiotic treatment [31,32,33]. In general, a higher diversity of gut microbiota can help ecosystems to maintain resilience after environmental stress by conferring redundancy on specific functions [34,35]. However, we also observed a significant decrease in microbiota evenness after gentamicin treatment. In this sense, the result could be another “dysbiosis” between infectious diarrhea therapy and antibiotic-induced microbiota disturbance. On the other hand, unlike jejunal microbes, which were significantly altered in terms of both diversity and structure, gentamicin did not cause significant changes in the ileal microbes. Considering the lack of changes in the total number of bacteria and *Escherichia coli* in the feces, we hypothesized that the treatment duration of gentamicin was not long enough to have a significant effect on the whole gut. However, these results revealed the potential negative effects of long-term use.

However, 16S rRNA cannot determine the taxonomic diversity of bacteria that do or do not respond to antibiotics, because its results include dormant, dead and quiescent bacteria [36,37,38,39]. Therefore, we used metabolomics to examine the metabolite profile of the jejunum, which would be more helpful for understanding the effects of antibiotics on the microbiota and host function than the microbial abundance differences alone [23]. Not surprisingly, the metabolite profile of the jejunum in the ETEC + Gen group, similar to its microbes, had a completely different compositional structure from the other two groups. This is consistent with the fact that the antibiotic-induced alterations of a few bacterial groups can cause major changes in gene and protein fluxes, regardless of whether similarities in the alpha diversity index exist [40].

To further evaluate the effect of gentamicin on metabolic processes, the Kyoto Encyclopedia of Genes and Genomes (KEGG) database was used to obtain the metabolic pathway information related to metabolites. Unsurprisingly, arachidonic acid metabolism was the most significantly downregulated pathway in the ETEC + Gen group, since it is a classic anti-inflammatory target [41]. Notably, the vancomycin resistance pathway was the most significantly upregulated pathway after gentamicin treatment, highlighting the danger that antibiotic exposure may promote the bacterial expression of antibiotic resistance [16]. This is consistent with a previous study’s conclusion that the most obvious response of the microbiota in the face of antibiotics is to first increase the acquisition and expression of a few genes that confer antibiotic resistance [42].

Given the strong association between metabolites and microbiota, the microbes were subjected to correlation analysis with metabolites showing significant changes within the pathway, while we used abundance change significance for further screening.

According to our results, *Bacteroides*, *uncultured_bacterium_f_Lachnospiraceae* and *uncultured_bacterium_f_Muribaculaceae* showed a significant increase in abundance and positive correlations with the vancomycin resistance pathway. Previous studies indicated that the relative abundance of *Bacteroides* correlated with a milk-oriented microbiome and showed remarkable decreases after weaning [43,44,45,46]. However, a previous study also indicated that *Lachospiraceae* genera began to emerge after weaning [47,48]. It is still debated whether the increase or decrease in species after weaning is an adaptation to stress or a factor related to diarrhea. There is also no consensus on the *Muribaculaceae* genera. It has been suggested that *Muribaculaceae* may be able to impede the colonization of *Clostridium perfringens* in the gut and provide protection against intestinal infections [49,50]. Conversely, by affecting the microbiota, typically including a decrease in *Muribaculaceae* abundance, this genera could improve bone health [51]. Although, here, we provide preliminary evidence for the threat of resistance and potential links between microbes and metabolism triggered by the use of gentamicin, it is difficult to determine if there is any causative effect in these correlations. There is still a long way to go in order to fully understand the impact of antibiotic use on the intestinal microbiota and metabolic pathways of piglets.

## 4. Materials and Methods

The experiments were conducted at the FengNing Swine Research Unit of China Agricultural University (Chengdejiuyun Agricultural and Livestock Co., Ltd., Hebei, China). All experimental procedures and animal care were approved by the China Agricultural University Animal Care and Use Committee (AW92902202-1-1, Beijing, China).

### 4.1. Piglet Selection

To be eligible for study participation, piglets (Duroc × Yorkshire × Landrace) had to meet the following inclusionary criteria: susceptibility to ETEC F4 infection; piglets born and weaned in the same batch; no history of intestinal diseases from birth; no history of drug treatments; similar body weight (BW); and equal numbers of barrows and gilts.

The ETEC-F4-susceptible piglets were selected on the basis of genotypes for F4ab/ac susceptibility [27]. DNA was extracted from blood obtained from the piglets 20 d after birth (venipuncture of the anterior vena cava) using a method described in the protocol supplied by the manufacturer (TIANamp Genomic DNA Kit, TIANGEN BIOTECH Co., Ltd., Beijing, China). The PCR of the MUC4 gene was performed using 3 min of initial denaturation at 95 °C, followed by 35 cycles of 98 °C for 10 s, 58 °C for 5 s and 68 °C for 5 s (LightCycler Real-Time PCR System, Roche, Germany). The primers are listed in Appendix A. The PCR product was digested with FastDigest XbaI (Thermo Fisher Scientific, Waltham, MA, USA) at 37 °C for 5 min. The size of the PCR product was 367 bp. The resistant samples were indigestible for XbaI, whereas the susceptible samples were digested into 151 bp and 216 bp fragments.

### 4.2. Bacterial Strain and Culture Conditions

The ETEC strain used in this study (ETEC K88, serotype O149:K91, K88ac) was purchased from the China Institute of Veterinary Drug Control (Beijing, China). The ETEC strain was grown under typical conditions in Luria–Bertani (LB) broth (Beijing AoBoXing Biotechnology Co., Ltd., Beijing, China) or on LB agar plates at 37 °C.

### 4.3. Experimental Design for Piglets

After piglet selection, all the piglets (42-day-old piglets) were susceptible to ETEC F4 and housed individually in stainless-steel holding crates (1.4 × 0.7 × 0.6 m). Feed and fresh water were available freely, with a controlled temperature (23 ± 2 °C) and humidity (55–65%). The diets (Appendix A) were formulated based on nutrient recommendations of the National Research Council [52].

As described above, thirty-six piglets with an average BW of 10.88 ± 0.13 kg were used in the therapeutic trial. The pigs were randomly assigned to one of three treatments: (1) the normal control group (NC) with vehicle (LB broth) challenge followed by placebo (normal saline) treatment; (2) the ETEC group, with ETEC challenge followed by placebo treatment; and (3) the ETEC + Gen group, with ETEC challenge followed by 2 mg/kg gentamicin (Gen) treatment (Figure 1A). The dose of the ETEC was determined in a pilot experiment. Body weight and feed intake were recorded at d 0 and d 5, and blood and fecal samples were taken on d 0, 24 h after ETEC challenge and on d 5. After sampling on the fifth day, the pigs (eight/group) were humanely killed via exsanguination after electrical stunning to obtain samples.

### 4.4. ELISA

The blood samples were maintained at room temperature for 2–3 h prior to centrifugation (3000× *g* for 10 min), and the serum was obtained and stored at −80 °C. The jejunum was homogenized with normal saline to obtain the supernatant. The levels of IL-1β, IL-8, IL-10, TNF-α, DAO and D-lactate were determined using ELISA kits (Nanjing Jiancheng Bioengineering Institute, Nanjing, China). The kit assays were carried out according to the protocol supplied by the manufacturer. The absorbance was read using a multimode microplate reader (450 nm, iMark, BIORAD, Hercules, CA, USA).

### 4.5. Fecal Indexes

The fecal fluid content was determined by desiccation in an oven (50 °C, 6 h) and weighed before and after desiccation [53]. The fecal score was defined as followed: 0, normal; 1, loose stool; 2, moderate diarrhea; and 3, severe diarrhea [54].

The quantification of the fecal microflora was performed via qRT-PCR, and the primers are listed in Appendix A. Briefly, stool DNA was extracted using a stool DNA kit (TIANGEN Biotech, Beijing, China) according to the manufacturer’s instructions. Specific standard curves were generated by constructing standard plasmids using a previously described method [55]. Specific PCR products were amplified from standard strains, purified using a Gel Extraction Kit (Beijing ComWin Biotech Co., Ltd., Beijing, China) and cloned into the pEASY-Blunt vector (TransGen Biotech, Beijing, China). After the verification of the sequence, the recombinant plasmid was isolated using the TIANprep Mini Plasmid Kit (TIANGEN Biotech, Beijing, China). The copy numbers of bacteria were calculated using the following formula: (6.0233 × 10^23^ copies/mol × DNA concentration (µg/µL))/(660 × 10^6^ × DNA size (bp)). A standard curve of the mean cycle threshold values against the logarithm of the template copy numbers was obtained via gradient dilution. For the detection of total bacteria, TB Green^TM^ Premix Ex Taq^TM^ Ⅱ (Takara Biotechnology Co. Ltd., Otsu, Shiga, Japan) and a LightCycler RT-PCR System (Roche, Germany) were used. For the detection of *E. coli*, a PrimerScript^TM^ PCR kit (Perfect Real Time; Takara) and LightCycler RT-PCR System (Roche, Germany) were used.

### 4.6. Histology

The jejunum and ileum sections were removed and washed in saline, fixed in 4% paraformaldehyde, and embedded in paraffin. After being cut into 4 μm thick slices, the tissue sections were stained with hematoxylin and eosin and then examined using a light microscope (Nikon Eclipse Ci, Japan). Photomicrographs were captured using a digital camera attached to the microscope (Nikon digital sight DS-FI2, Japan).

Histological damage was quantitatively assessed as described in Appendix A. The sum of four subscores resulted in a combined score ranging from 0 (no changes) to 12 (widespread cellular infiltrates and extensive tissue damage). Each section in each group was selected and photographed at a 40 × field of view. Five complete villi were selected from each section, and the villus length (μm) and crypt depth (μm) were measured.

### 4.7. Microbiota Composition Assessed via 16S rRNA Sequencing Analysis

Jejunal mucosal samples were obtained for sequencing. Genomic DNA was extracted using a TGuide S96 Magnetic Soil/Stool DNA Kit (Tiangen Biotech (Beijing) Co., Ltd.). The V3–V4 regions of the bacterial 16S rRNA gene were amplified with the universal primers 338F (5′-ACTCCTACGGGAGGCAGCAG-3′) and 806R (5′-GGACTACHVGGGTWTCTAAT-3′) [56]. The PCR amplicons were purified with Agencourt AMPure XP Beads (Beckman Coulter, Indianapolis, IN, USA) and quantified using a Qubit dsDNA HS Assay Kit and Qubit 4.0 Fluorometer (Invitrogen, Thermo Fisher Scientific, Eugene, OR, USA). The purified amplicons were pooled in equimolar amounts and sequenced on an Illumina NovaSeq 6000 (Illumina, Santiago CA, USA). The raw data were primarily filtered using Trimmomatic (version 0.33) [57]. The identification and removal of the primer sequences were performed using Cutadapt (version 1.9.1) [58]. Use USEARCH (version 10) assembly and UCHIME (version 8.1) were used to remove chimeras [59,60]. The operational taxonomic units (OTUs) were clustered with 97% similarity using USEARCH (v10.0) [57]. The taxonomy annotation of the OTUs was performed based on the Naive Bayes classifier in QIIME2 using the SILVA database (release 132) with a confidence threshold of 70% [60,61].

### 4.8. LC-MS/MS Analysis

The jejunum tissues were collected for metabolomics analysis. Metabolites with variable importance in projection (VIP) of ≥1 and *p* < 0.05 were regarded as statistically significant (differentially expressed metabolites). Pathway analysis was performed by integrating a hypergeometric test and topology analysis.

The LC/MS system for metabolomics analysis was composed of a Waters Acquity I-Class PLUS ultra-high-performance liquid tandem Waters Xevo G2-XS QTof high-resolution mass spectrometer. The column used was purchased from Waters Acquity, being a UPLC HSS T3 column (1.8 µm, 2.1 mm × 100 mm). For the positive ion mode, the conditions were as follows: mobile phase A: 0.1% formic acid aqueous solution; mobile phase B: 0.1% formic acid acetonitrile. For the negative ion mode, the conditions were as follows: mobile phase A: 0.1% formic acid aqueous solution; mobile phase B: 0.1% formic acid acetonitrile. The injection volume was 1 μL.

A Waters Xevo G2-XS QTOF high-resolution mass spectrometer can collect primary and secondary mass spectrometry data in the MSe mode under the control of the acquisition software (MassLynx V4.2, Waters). In each data acquisition cycle, dual-channel data acquisition can be performed with both low collision energy and high collision energy at the same time. The low collision energy is 2 V, the high collision energy range is 10–40 V, and the scanning frequency is 0.2 s for a mass spectrum. The parameters of the ESI ion source were as follows: capillary voltage, 2000 V (positive ion mode) or −1500 V (negative ion mode); cone voltage, 30 V; ion source temperature, 150 °C; desolvent gas temperature, 500 °C; backflush gas flow rate, 50 L/h; desolventizing gas flow rate, 800 L/h.

The raw data, collected using MassLynx V4.2, were processed using Progenesis QI software for peak extraction, peak alignment and other data processing operations based on the Progenesis QI software online METLIN database and Biomark’s self-built library for identification. Theoretical fragment identification and mass deviation were within 100 ppm.

The original peak area information was normalized to the total peak area, and principal component analysis and Spearman’s correlation analysis were used to judge the repeatability of the samples within the group and the quality control samples.

### 4.9. Statistical Analysis

A statistical analysis was performed using Prism software (GraphPad 9.0.1). The data were first checked for a normal distribution and plotted in the figures as the mean ± SEM. For experiments containing more than two relative groups, one-way ANOVA followed by Dunnett’s or Tukey’s multiple comparisons post hoc test was performed. Differences in the bacterial data were evaluated using the Wilcoxon rank sum test or Kruskal-Wallis H test. PICRUSt2 software was used to compare the species composition information obtained from the 16S sequencing data in order to infer the functional gene composition. Random forest analysis was performed using R v3.1.1 (random forest v4.6-10) to obtain key species that had important effects on intergroup differences. Python 2.7.8 (scipy-0.14.1) and Spearman’s analysis were used to calculate the correlation coefficient between the species and phenotypes, which were visualized in the form of a heatmap, and *p* values < 0.05 were considered statistically significant.

## 5. Conclusions

The present study demonstrated that dramatic changes are produced on both the jejunal microbial and metabolic levels in piglets shortly after gentamicin treatment. The jejunal microbiota showed remarkable structural differences accompanied by a decrease in evenness, despite the increase in diversity. The metabolomic approach showed that gentamicin significantly increased the enrichment of vancomycin resistance pathways, among which Bacteroides, *uncultured_bacterium_f_Lachnospiraceae* and *uncultured_bacterium_f_Muribaculaceae* were the most prominent and most worthy of attention.

In summary, the bacterial evenness of the jejunal lumen was diminished after gentamicin administration. The alterations in the abundance and composition of gut microbiota and metabolites both support the dysfunction of gut microbiota. Over the past few years, studies have revealed remarkable interactions between microbiota and antibiotic treatments. This report extends these connections by demonstrating the effects of altered gut microbiota and the metabolic profiles of gentamicin. We hope that a further understanding of the roles of these processes will facilitate the conception and realization of new therapeutic and preventive strategies.

## Figures and Tables

**Figure 1 antibiotics-12-01349-f001:**
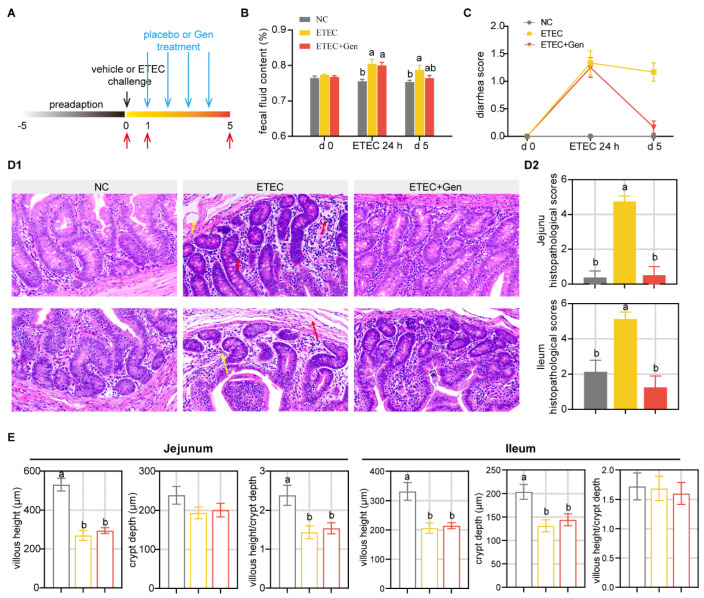
The effect of gentamicin on the clinical symptoms and intestinal morphology of ETEC-challenged piglets. (**A**) experimental designs. (**B**) Fecal fluid content analysis. (**C**) Diarrhea score changes. Scoring standards: 0, normal; 1, loose stool; 2, moderate diarrhea; 3, severe diarrhea. n = 12, mean ± SEM. (**D1**) Representative images of jejunum (up) and ileum (down) withby H&E staining (200×) and (**D2**) histopathological scores. The red arrow indicates inflammatory cell infiltration in the mucosal layer, and the yellow arrow indicates edema status in the submucosa. n = 8, mean ± SEM. (**E**) Morphological measurement of jejunum and ileum. n = 8, mean ± SEM. ANOVA followed by Tukey’s multiple comparisons test. Different lowercase letters within each group indicate significantly different values (*p* < 0.05).

**Figure 2 antibiotics-12-01349-f002:**
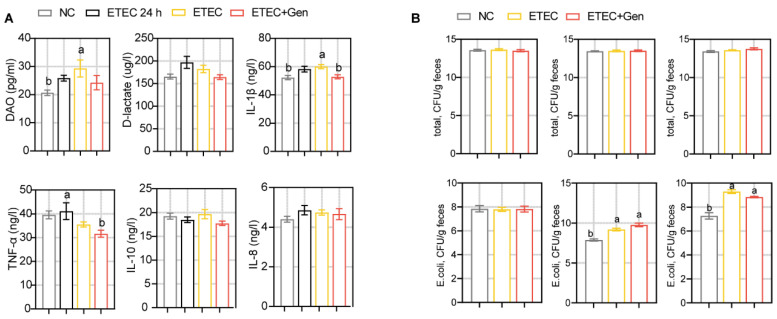
The effect of gentamicin on cytokines and fecal bacteria in ETEC-challenged piglets. (**A**) Cytokines’ measurement in serum. (**B**) Measurement of bacteria in feces. n = 8, mean ± SEM. ANOVA followed by Tukey’s multiple comparisons test. Different lowercase letters within each group indicate significantly different values (*p* < 0.05).

**Figure 3 antibiotics-12-01349-f003:**
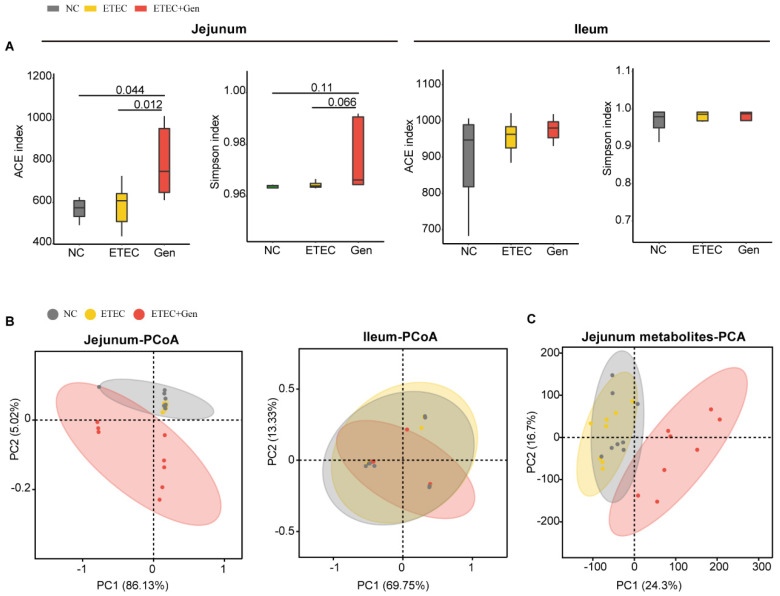
Gentamicin treatment resulted in significant variation in jejunal microbiota and metabolites in ETEC-challenged piglets. (**A**) Alpha diversity comparisons of microbial communities. n = 8, mean ± SEM. Numbers on the horizontal lines indicate p values using paired Student’s *t* test. (**B**) PCoA of 16S genes. Using an OTU definition of 97% similarity, based on Bray_Curtis. (**C**) Principal component analysis of untargeted metabolomics data for jejunal tissues. Ellipses represent the 95% confidence interval.

**Figure 4 antibiotics-12-01349-f004:**
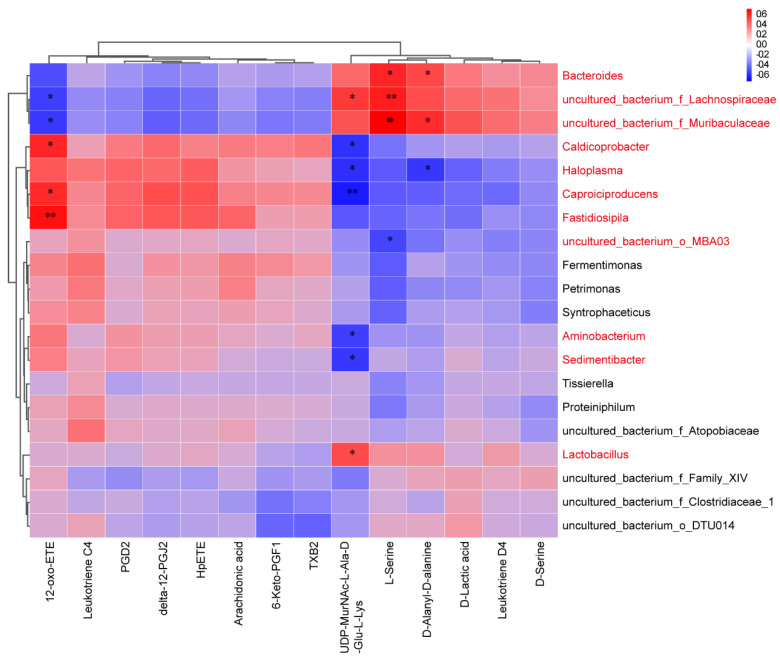
Correlation matrix between specific metabolites and microbiota (spearman correlation, *, *p* < 0.05, **, *p* < 0.01).

**Figure 5 antibiotics-12-01349-f005:**
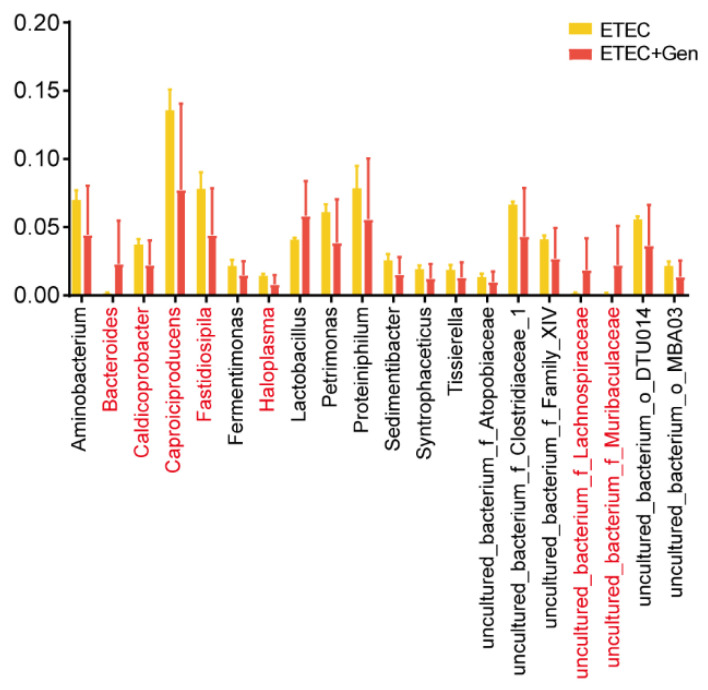
Screening of microorganisms. Graph showing the selected microorganisms that differed significantly between the ETEC-challenged and gentamicin-treated piglets.

## Data Availability

Not applicable.

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
