# Peer review of "Multi-Omics Analysis Reveals the Gut Microbiota Characteristics of Diarrheal Piglets Treated with Gentamicin"

_antibiotics, 2023, doi:10.3390/antibiotics12091349_

Round 1

Reviewer 1 Report

 Title: “Multi-omics analysis reveals gut microbiota characteristics in diarrheal piglets treated with gentamicin”

Several changes have been suggested and incorporated in the article as track changes especially discussion needs improvement as very less reference are cited.

Note: Please refer to the attached file for specific comments and suggested improvements.

Author Response

We would like to thank the editor and reviewers for your professional comments and suggestions, which are very helpful in improving our manuscript. Based on the instructions provided in your letter, we uploaded the file of the revised manuscript. Accordingly, we have uploaded a copy of the original manuscript with all the changes highlighted by using the track changes mode in MS Word. Appended to this letter is our point-by-point response to the comments raised by the reviewers. The comments are reproduced and our responses are given directly afterward in a different color (red). We would like also to thank you for allowing us to resubmit a revised copy of the manuscript.

Response to comment

First of all, we would like to thank you for your extensive English revision (especially in the use of “we”), which is very helpful for the improvement of our article level. Thank you so much for your careful check. We all accept and learn a lot. Followed is our point-by-point response to your comments:

Suggestions 1: Explain the taxonomic variation at different levels (phylum order genus etc) observed in different groups.

Reply: Thank you for your nice advice. We agree with the suggestion that taxonomic variation at different levels is important for the analysis of bacterial diversity. Since subsequent screening was also performed with the use of between-group differences, the added analysis was shown in Supplemental table 5 and 6.

Suggestions 2: Insert these reference

Reply: We gratefully thanks for the precious time the reviewer spent making these remarks. We've read the literatures, cited the literatures to the appropriate places and learned a lot.

Suggestions 3: Rephrase the senteence

Reply: We feel sorry for the vague statement. Follow your suggestion, we revised the manuscript “Given the strong association between metabolites and microbiota, the microbes were subjected to correlation analysis with metabolites with significant changes within the pathway, while using abundance change significance for further screening”.

Suggestions 4: Condition speed time?

Reply: Thank you so much for your careful check. We are very sorry for our negligence of the statement. We revised the manuscript “Blood samples were maintained at room temperature for 2-3 h prior to centrifugation (3000× g for 10 min), and the serum was obtained and stored at -80 ℃.”

Suggestions 5: Trimming criteria for clean reads?

Reply: Thank you for your nice advice. We statement in manuscript “Raw data were primarily filtered by Trimmomatic (version 0.33)”. Trimmomatic is a trimming tool designed for Illumina sequencing reads. It is applicable to both SE and PE reads using either phred33 or phred64 quality score. For SE reads treatment, an input file and output file name and corresponding parameters are required. For PE reads treatment, forward and reverse fastq files need to be provided. Parameter setting: Window size was set as 50 bp. The reads will be cut from the start of the window once average Q-score within the window is lower than 20.

We tried our best to improve the manuscript and made some changes in the manuscript. These changes will not influence the content and framework of the paper. We appreciate for Editors/Reviewers’ warm work earnestly, and hope that the correction will meet with approval. Once again, thank you very much for your comments and suggestions.

Reviewer 2 Report

Abstract:

It should contain the main objectives of the study before presentation of the results.

Keywords:

Remove the bold format of "gentamicin" 

Results: 

Authors should present the intestinal morphometry data and with proper statistical analysis.

Lines 72 to 74:  Realocate this paragraph to discussion section.

Lines 74 to 76: Authors need to improve the mention about the growth performance, diarrhea evaluation and intestinal morphometry in Results section, highlighting the main findings and providing at least the means ± SD or SEM (P-value) in this section.

Fig. 1B and 1C are not related to the fecal fluid content and diarrhea score. Authors should correct it in text. Also, figures C and D, as well as, Figures 2A, 2B and 3A, 3B were not provided in the supplementary files. Maybe authors made a mistake submiting the same file twice as non-published data and as supplementary data.

Why authors have decided to evaluate only the jejunum and not also the duodenum, since it is the main absorption site of small intestine?

Discussion:

Authors should improve the discussion about the piglets growth performance and provide references that corroborate or not the results found in their study. The same for the intestinal morphometry data, for diarrhea findings, DAO, lactate and cytokine levels.

Materials and methods:

Authors should improve the text of "4.1 Pig selection" with the answers for the questions below:

What was the breed of pigltes (father line x mother line)?

What was the average age of piglets used in the study? 

The piglets blood were collected from which via?

4.3 Experimental design for piglets:

What was the vehicle used in treatments?

ELISA:

What was the wavelength used for the readings?

Histology:

Line 273: correct the mention of variable in text (is it villi height?)

Why authors decided to measure only 10 villi height and 10 crypt depth on 5 fields of view instead of increasing the number of measurements for each variable (villi height and crypt depth)? Provide a reference for the protocol used.

Supplementary files:

Authors should present the intestinal morphometry data and with proper statistical analysis. Supplementary table 3 cited in text was not provided.

Fig. 1B and 1C are not related to the fecal fluid content and diarrhea score. Authors should correct it in text.

Also, figures C and D, as well as, figures 2A, 2B and 3A, 3B were not provided in the supplementary files. 

Abstract:

Line 11: Please correct "theatment" to "treatment".

Author Response

We would like to thank the editor and reviewers for your professional comments and suggestions, which are very helpful in improving our manuscript. Based on the instructions provided in your letter, we uploaded the file of the revised manuscript. Accordingly, we have uploaded a copy of the original manuscript with all the changes highlighted by using the track changes mode in MS Word. Appended to this letter is our point-by-point response to the comments raised by the reviewers. The comments are reproduced and our responses are given directly afterward in a different color (red). We would like also to thank you for allowing us to resubmit a revised copy of the manuscript.

Response to comment

Abstract:

It should contain the main objectives of the study before presentation of the results.

Reply: We gratefully appreciate for your nice comment. We added the following statement in abstract: “However, a clear picture of the influences of gentamicin, which is employed for the treatment of bacterial diarrhea in animal production, is largely unknown. Here, we addressed it using enterotoxigenic Escherichia coli (ETEC) F4 susceptible piglet models treated with gentamicin.”

Keywords:

Remove the bold format of "gentamicin"

Results:

Authors should present the intestinal morphometry data and with proper statistical analysis.

Fig. 1B and 1C are not related to the fecal fluid content and diarrhea score. Authors should correct it in text. Also, figures C and D, as well as, Figures 2A, 2B and 3A, 3B were not provided in the supplementary files. Maybe authors made a mistake submiting the same file twice as non-published data and as supplementary data.

Supplementary files:

Authors should present the intestinal morphometry data and with proper statistical analysis. Supplementary table 3 cited in text was not provided.

Fig. 1B and 1C are not related to the fecal fluid content and diarrhea score. Authors should correct it in text.

Also, figures C and D, as well as, figures 2A, 2B and 3A, 3B were not provided in the supplementary files.

Reply: We gratefully thanks for the precious time the reviewer spent making careful check. The questions above could be replied together. We re-checked the manuscript and found that the text and figures were normal. We guess whether it is due to the problem of typesetting or web page uploading, so we will upload again after this check, hoping to answer the above questions. If there are still problems, please do not hesitate to contact us, and we will try to solve them by providing separate figure information or other measures.

Lines 72 to 74: Realocate this paragraph to discussion section.

Reply: Thank you for your nice advice. We agree with the suggestion that this type of description is more appropriate for the discussion section. However, a similar description is already available in the discussion section (the second paragraph of the discussion). If you think it should be removed, we will make further modifications.

Lines 74 to 76: Authors need to improve the mention about the growth performance, diarrhea evaluation and intestinal morphometry in Results section, highlighting the main findings and providing at least the means ± SD or SEM (P-value) in this section.

Reply: We feel sorry for the vague statement. Follow your suggestion, we revised the manuscript “After gentamicin treatment, the growth performance of infected piglets was numerically improved, although not significantly (Supplemental Table 4). Similarly, a significantly improvement was observed in fecal fluid content and diarrhea score (Fig.1 B and C). Histological analysis showed obvious impairment in the jejunum of ETEC-challenged piglets as reflected by severely damaged mucosal structures and substantial inflammatory cell infiltration (Fig.1D).”

Why authors have decided to evaluate only the jejunum and not also the duodenum, since it is the main absorption site of small intestine?

Reply: F4 positive ETEC are observed adhering to most of the jejunum and ileum’s enterocyte brush border membrane of intestinal mucosa [1, 2], Therefore, the jejunum and ileum became the main object of our study. However, the effect of gentamicin treatment on the ileum was minimal (data not shown), so we mainly analyzed the part of the jejunum in the article.

Discussion:

Authors should improve the discussion about the piglets growth performance and provide references that corroborate or not the results found in their study. The same for the intestinal morphometry data, for diarrhea findings, DAO, lactate and cytokine levels.

Reply: Thank you so much for your careful check. We are very sorry for our negligence of the statement in the limitation of these symptoms. Follow your suggestion, we revised the manuscript “Although there was no significant improvement in growth performance, consistent with other in-feed antibiotics in previous studies, gentamicin treatment significantly alleviated symptoms of diarrhea and intestinal injury [28-30]. These were also verified by significantly decreased DAO, IL-1β and TNF-α levels in serum. The failure to significantly improve growth performance may be due to the inadequate duration of treatment in the present study.”

Materials and methods:

Authors should improve the text of "4.1 Pig selection" with the answers for the questions below:

What was the breed of pigltes (father line x mother line)?

What was the average age of piglets used in the study?

The piglets blood were collected from which via?

4.3 Experimental design for piglets:

What was the vehicle used in treatments?

ELISA:

What was the wavelength used for the readings?

Reply: Thank you so much for your careful check. Below is our reply and we have made the corresponding changes in the article:

Piglets (Duroc × Yorkshire × Landrace, 42-day-old)

Blood samples: venipuncture of the anterior vena cava.

Vehicle: LB broth

Wavelength: 450 nm

Why authors decided to measure only 10 villi height and 10 crypt depth on 5 fields of view instead of increasing the number of measurements for each variable (villi height and crypt depth)? Provide a reference for the protocol used.

Reply: Thank you so much for your careful check. We are very sorry for our mistakes of the statement in Histology. The previous description was used in other studies. After correction: “Each section in each group was selected and photographed at 40 × field of view. Five complete villi were selected from each section, and the villus length (μm) and crypt depth (μm) were measured.”

We tried our best to improve the manuscript and made some changes in the manuscript. These changes will not influence the content and framework of the paper. We appreciate for Editors/Reviewers’ warm work earnestly, and hope that the correction will meet with approval. Once again, thank you very much for your comments and suggestions.

[1] Fairbrother JM, Gyles CL. Colibacillosis. In: Zimmerman JJ, Karriker LA, Ramirez A, Schwartz KJ, Stevenson GW, editors. Disease of Swine. 10th ed; 2012. p. 723–47.

[2] Luppi A. Swine enteric colibacillosis: diagnosis, therapy and antimicrobial resistance. Porcine Health Manag. 2017 Aug 8;3:16. doi: 10.1186/s40813-017-0063-4. PMID: 28794894; PMCID: PMC5547460.

Round 2

Reviewer 1 Report

Manuscript is improved now

Reviewer 2 Report

Authors have answered all questions and made the requested corrections. Therefore, the final decision regarding the article can be made by the Editor.